# Microstructure and Electrical Conductivity of Electrospun Titanium Oxynitride Carbon Composite Nanofibers

**DOI:** 10.3390/nano12132177

**Published:** 2022-06-24

**Authors:** Gorazd Koderman Podboršek, Špela Zupančič, Rok Kaufman, Angelja Kjara Surca, Aleš Marsel, Andraž Pavlišič, Nejc Hodnik, Goran Dražić, Marjan Bele

**Affiliations:** 1Department of Materials Chemistry, National Institute of Chemistry, Hajdrihova 19, SI-1000 Ljubljana, Slovenia; angelja.k.surca@ki.si (A.K.S.); ales.marsel@ki.si (A.M.); nejc.hodnik@ki.si (N.H.); goran.drazic@ki.si (G.D.); 2Jožef Stefan International Postgraduate School, Jamova 39, SI-1000 Ljubljana, Slovenia; 3Department of Pharmaceutical Technology, Faculty of Pharmacy, University of Ljubljana, Aškerčeva 7, SI-1000 Ljubljana, Slovenia; spela.zupancic@ffa.uni-lj.si; 4Faculty of Mathematics and Physics, University of Ljubljana, Jadranska 19, SI-1000 Ljubljana, Slovenia; rok.kaufman@student.fmf.uni-lj.si; 5Department of Catalysis and Chemical Reaction Engineering, National Institute of Chemistry, Hajdrihova 19, SI-1000 Ljubljana, Slovenia; andraz.pavlisic@ki.si

**Keywords:** titanium oxynitride, carbon, composite, nanofibers, in situ four-probe electrical conductivity, sheet resistivity, specific capacitance, electrospinning

## Abstract

Titanium oxynitride carbon composite nanofibers (TiON/C-CNFs) were synthesised with electrospinning and subsequent heat treatment in ammonia gas. In situ four-probe electrical conductivity measurements of individual TiON/C-CNFs were performed. Additionally, the TiON/C-CNFs were thoroughly analysed with various techniques, such as X-ray and electron diffractions, electron microscopies and spectroscopies, thermogravimetric analysis and chemical analysis to determine the crystal structure, morphology, chemical composition, and N/O at. ratio. It was found that nanofibers were composed of 2–5 nm sized titanium oxynitride (TiON) nanoparticles embedded in an amorphous carbon matrix with a small degree of porosity. The average electrical conductivity of a single TiON/C-CNF was 1.2 kS/m and the bulk electrical conductivity of the TiON/C-CNF fabric was 0.053 kS/m. From the available data, the mesh density of the TiON/C-CNF fabric was estimated to have a characteristic length of 1.0 µm and electrical conductivity of a single TiON/C-CNF was estimated to be from 0.45 kS/m to 19 kS/m. The electrical conductivity of the measured TiON/C-CNFs is better than that of amorphous carbon nanofibers and has ohmic behaviour, which indicates that it can effectively serve as a new type of support material for electrocatalysts, batteries, sensors or supercapacitors.

## 1. Introduction

Titanium oxynitride (TiON) is a material that has gained a lot of attention in recent years, due to its versatile properties and is, therefore, being researched in many different scientific fields, such as optoelectronics [1], water electrolysis [2], fuel cells [3], supercapacitors [4,5,6], and energy storage [7,8,9,10]. TiON is a general term describing a group of compounds with the formula TiO_x_N_y_ that can have many remarkable properties of both TiO_2_ (high chemical stability) and TiN (high electrical conductivity). Moreover, its properties can even be fine-tuned by varying the N/O at. ratio [11]. For this reason, it is considered as a promising alternative to carbon-based supports for electrocatalysts in electrolysers, as demonstrated by the research of our group [12,13,14].

For such applications, large surface area and fast charge transfer are highly desirable properties. Here, 1D nanomaterials, such as nanowires and nanofibers, show their structural advantages. Due to their low aspect ratio (d_min_/d_max_), the conductivity is much higher compared to other types of nanomaterials, where there are more phase boundaries. Because of their nanoscale size, the 1D nanomaterials still have a very high surface area where the electrocatalyst nanoparticles can be dispersed and stay electronically well connected. Nanofibers are used in many different fields, such as supercapacitors, biosensors, tissue engineering, photovoltaics, fuel cells, hydrogen storage, water filters, and more [15].

The development of storage devices for renewable energy has been one of the main focuses for green energy research for years. Most of the research can be grouped into the following three major fields: batteries, fuel cells, and supercapacitors. Supercapacitors are interesting because they can provide a higher power density than batteries [16]. They also have a fast charge-discharge time and good energy density. One branch of promising materials for supercapacitors are metal oxides, due to their excellent capacitance, which is usually higher than materials that are based on carbon. However, metal oxides have poor stability and electrical conductivity [17].

One of the most important properties of nanofibers and nanowires for electrocatalytic applications is their electrical conductivity. There are many different methods to measure the electrical conductivity of nanomaterials and specifically nanowires and nanofibers. They can be divided into different categories according to the type of electrical measurement, which are as follows: two-probe electrical conductivity [18], four-probe electrical conductivity [18,19], impedance [20], and Hall effect [7]. Another way to categorise electrical conductivity measurements is by sample size, including bulk measurement [19], single nanofiber/nanowire [8,21], and array of nanofibers/nanowires [22].

Four-probe electrical conductivity is necessary to accurately measure the intrinsic electrical conductivity of the material at the nanoscale in single nanofibers/nanowires, without the influence of nanocontact resistance that cannot be excluded by a two-probe technique [18]. Measuring the electrical conductivity of single nanofibers/nanowires is advantageous to bulk electrical conductivity measurements because we can avoid the contact resistance between nanofibers/nanowires and measure the intrinsic electrical conductivity of the material.

Many types of nanofibers/nanowires have been synthesised and their electrical conductivity determined, such as nanofibers made from pure elements [22], polymer (poly(3,4-ethylenedioxy- thiophene) (PEDOT) [18]), polymer-TiO_2_ hybrid (poly (methyl methacrylate) (PMMA)/TiO_2_ [23]), TiO_2_ based (TiO_2_ [24], TiO_2_N_x_ [20]), TiN based (TiN [25], TiN/Cu [26]), C [27], C doped with N [28], and C hybrids (TiO_2_/TiON/C [19], TiC/C [29]).

Other types of TiON nanomaterials have already been synthesised and their electrical conductivity has been determined, such as thin films [1,2,11], nanoparticles [3,4], nanorods [5,6], and nanostructures [10]. Electrical conductivity and other electrical properties of 1D TiON nanomaterials have been measured and reported [7,8,9]. Chen, T. T. et al. synthesised flexible porous TiON sheets consisting of TiON nanofibers via a two-step process (hydrothermal reaction and nitridation) for their use as an electrochemical electrode and potentially in supercapacitors. They measured the bulk conductivity of the TiON sheets using the Hall effect measurement system and reported conductivities from 1.66 to 31.45 kS/m [7]. Dong, J. et al. synthesised TiON mesoporous nanowires by nitridating hydrogen titanate nanowires for their use as an anode in sodium hybrid electrochemical capacitors. They measured the single nanowire conductivity via a two-probe conductivity measurement and reported 3000 kS/m [8]. Noh, Y. et al. synthesised 1D TiON nanowires by electrospinning and nitridation for the use in capacitive charge storage and electroreduction of oxygen. They measured the electrical properties of the bulk sample with impedance spectroscopy and found that the carbon species increase the conductivity of TiON nanowire [9].

In this study, TiON carbon composite nanofibers (TiON/C-CNFs) were synthesised by electrospinning and subsequent heat treatment in ammonia gas. The obtained nanofibers were thoroughly investigated with X-ray diffraction (XRD), selected area electron diffraction (SAED), scanning transmission electron microscopy (STEM), scanning electron microscopy (SEM), electron energy loss spectroscopy (EELS), CHNS elemental analysis, thermogravimetric analysis (TGA), SEM energy dispersive X-ray spectroscopy (SEM EDXS), STEM energy dispersive X-ray spectroscopy (STEM EDXS), and Raman spectroscopy to determine the crystal structure, morphology, chemical composition, and N/O at. ratio. In situ four-probe electrical conductivity of single TiON/C-CNFs was measured. Sheet resistance of TiON/C-CNF fabric was also measured with the van der Pauw method and mesh density was estimated from these two electrical measurements. Experimental electrical conductivity results were compared to calculated values using a two-phase composite model and obtained fit was relatively good. It was shown that electrospun TiON/C-CNFs seem to be a promising candidate for a high capacitance, high conductivity supercapacitor.

## 2. Materials and Methods

### 2.1. Preparation of TiON/C-CNF Fabrics

To prepare TiON/C-CNF fabrics for sheet resistivity and capacitance measurements, a procedure was chosen that gives more monodisperse TiON/C-CNFs. In this way, the sheet resistivity results could be more easily compared to the theoretical model. A polymer solution was composed of 0.6 g of Polyvinylpyrrolidone F90 (PVP, BASF, Ludwigshafen, Germany), 1.0 g of titanium (IV) isopropoxide (TIP, 97%, Merck KgaA, Darmstadt, Germany), 2.0 g of glacial acetic acid (acetic acid, Merck KgaA, Darmstadt, Germany) and 6.4 g of ethanol absolute (ethanol, Merck KgaA, Darmstadt, Germany). First, PVP was dissolved in ethanol and then the rest of the components were added. The nanofibers were produced using electrospinning with a Spinbox system (BioInicia, Valencia, Spain). The syringe with polymer was placed in the pump, which generated a constant flow rate of 800 μL/h through the needle. Positive high voltage of 14.5 kV was applied to the needle that was positioned 15 cm from the nozzle. Electrospinning was performed at 26 °C and 50% relative humidity. The sample name TiON/C-CNF1 will be used where this type of TiON/C-CNFs was used (1st sample).

For in situ four-probe conductivity measurements, microstructure, and composition analysis, a procedure was chosen that gives more polydisperse TiON nanofabrics. In this way, suitable single TiON/C-CNFs could be chosen more easily. A polymer solution for the preparation of TiON/C-CNFs was composed of 0.8 g of PVP, 1.7 g of TIP, 3.3 g of acetic acid, and 4.2 g of ethanol. First, PVP was dissolved in ethanol and then the rest of the components were added. A syringe pump (model R-99E, RazelTM, Linari Engineering, Pisa, Italy) was used to feed a polymer solution to the needle, with a constant rate of 763 μL/h. A high voltage of 10.0 kV at the needle was achieved by connection to a voltage generator (model HVG-P60-R-EU, Linari Engineering, Pisa, Italy). Positive high voltage of 10.0 kV was applied to the needle that was positioned 15 cm from the nozzle. Electrospinning was performed at 23 °C and 21% relative humidity. The sample name TiON/C-CNF2 will be used where this type of TiON/C-CNFs was used (2nd sample).

We were not able to avoid slight differences in composition due to the aim of synthesising two samples with different diameter size dispersions.

Electronic conductivity of the prepared nanofibers was accomplished with thermal treatment in an ammonia atmosphere. The temperature was first increased at a rate of 2 °C/min to 250 °C for 1 h, then at a rate of 5 °C/min to 700 °C for 2 h, and then cooled to room temperature with a rate of 5 °C/min. The flow of ammonia gas was kept constant at 50 cm^3^/min.

### 2.2. Preparation of Single TiON/C-CNF2 Samples for Electrical Measurements

A small piece of the synthesised nanofiber fabric was put in a few mL of MiliQ water and broken up into tinier pieces with tweezers and vigorous mixing. A few 100 mL of MiliQ water was added and the suspension was mixed. The suspension was left for a few minutes to settle down. The surplus water was decanted to remove smaller pieces of nanofibers and potential soluble remnants from the synthesis. An amount of 2 µL of the suspension was drawn with a pipette and drop coated on the Protochips Electrical E-chip (finger configuration—fusion select, Protochips inc., Morrisville, NC, USA) as close to the contacts as possible. The drop was left to dry. The platinum contacts on selected nanofibers were made in the Helios NanoLab 650i Dual-beam focused ion beam (FIB)SEM (FIB-SEM) system (Thermo Fisher Scientific, Waltham, MA, USA., previously FEI Company). The E-chip with wired nanofiber was put on Protochips Fusion select holder (Protochips inc., Morrisville, NC, USA), inserted into a transmission electron microscope (TEM), and connected to Protochips Aduro^TM^ power supply (Protochips inc., Morrisville, NC, USA) for in situ four-probe electrical measurements.

### 2.3. Electrical Measurements of TiON/C-CNF1 Fabric

Sheet resistivity of TiON/C-CNF1 fabric samples was measured with the van der Pauw method [30,31], which allows accurate measurements of sheet resistivity of a sample of arbitrary solid shape. Samples of a roughly square shape of approximate dimensions 10 mm × 10 mm × 30 µm were attached to a printed circuit board with a conductive silver paste. The printed circuit board itself is square shaped with pins soldered to copper in all four corners. This allows for a robust connection of the fragile samples to the measuring equipment. The measurements were done with Multichannel Potentiostat BioLogic VMP3 (Biologic, Seyssinet-Pariset, France) and repeated in reciprocal configuration and with reversed polarity to confirm the repeatability of measurements.

### 2.4. Electrochemical Measurements of TiON/C-CNF1 Fabric

The sample (10 mm x 10 mm x 30 µm) was put onto a gas diffusion layer (GDL, Spectracarb 2050A-1050, FuelCellStore, College Station, TX, USA) for structural support and a Nafion 117 membrane (FuelCellStore, College Station, TX, USA) was put on top of that, to ensure a good proton flow and to prevent leakage from the cell. As the electrolyte, 4 M HClO_4_ (Suprapur, Carl Roth GmbH + Co. KG, Karlsruhe, Germany) was used. Pt mesh counter electrode was used. The electrochemical impedance spectroscopy (EIS) was performed with a PARSTAT 2273 potentiostat (Advanced Measurement Technology inc., Oak Ridge, TN, USA) from 1 MHz to 0.1 mHz. The cyclic voltammetry (CV) was measured from −0.15 V to 0.9 V at a scan rate of 10 mV/s. The charge–discharge curves were obtained using a Biologic SP-300 potentiostat (Biologic, Seyssinet-Pariset, France) from 0.05–0.9 V at constant currents (4 Ag^−1^, 10 Ag^−1^, 20 Ag^−1^, 40 Ag^−1^, 60 Ag^−1^). The mass of the sample was determined by weighing a 1 cm^2^ sheet and then calculating the mass for the area of the sample that was exposed to the electrolyte (0.07 cm^2^). The areal mass loading was 1 mg/cm^2^.

### 2.5. STEM, EELS and STEM EDXS Analysis

STEM imaging was performed on TiON/C-CNF2s in a probe Cs-corrected TEM ARM Jeol 200 CF (JEOL Ltd., Tokyo, Japan), equipped with a solid state detector (SSD) JEOL energy-dispersive X-ray (EDX) spectrometer (JEOL Ltd., Tokyo, Japan) and a GATAN Quantum ER dual- EELS system (Gatan Inc., Pleasanton, CA, USA). An operational voltage of 80 kV was employed. The images were taken in STEM high angle annular dark field(STEM-HAADF) mode, using probe size 6C and the effective camera length was 8 cm. Probe size 6C was used for EELS analysis, effective camera length was 3 cm, probe size 2C was used for STEM EDXS analysis and the effective camera length was 8 cm. STEM-HAADF imaging angles were 68–175 mrad.

### 2.6. Sample Preparation of TiON/C-CNF2s for STEM, EELS and STEM EDXS Analysis

The sample was prepared in two ways. To obtain individual TiON/C-CNFs, a small piece of the synthesised nanofiber fabric was put on a TEM grid (lacey carbon on copper, mesh 300, Structure Probe Inc., West Chester, PA, USA) and rubbed onto it with a toothpick. Excess material was blown away with compressed air. To obtain a FIB lamella of the cross-sections of TiON/C-CNFs, 2 µL of the suspension (prepared in the same way as for electrical measurements) were drawn with a pipette, drop coated on a piece of silicon wafer, and left to dry. The prepared sample was put into the FIB-SEM. An appropriate location on the TiON/C-CNFs fabric was chosen so that a couple dozen of TiON/C-CNFs were aligned more or less parallel at this point/location. A thick layer of platinum was put on top of the TiON/C-CNFs and a lamella was cut out with Ga ion sputtering. The lamella was transferred to a lift-out TEM grid (Omniprobe, 3 post, copper, Structure Probe Inc., West Chester, PA, USA) and thinned additionally with Ga ion sputtering.

### 2.7. Raman Spectroscopy

Raman spectra of TiON/C-CNF2s were obtained using a confocal WITec alpha 300 spectrometer (WITec GmbH, Ulm, Germany). The spectra were recorded with a green 532 nm laser excitation wavelength using 100 scans and an integration time of 1 s. The spectra were recorded on two different sites. At each site, four spectra were measured subsequently using the increasing laser powers of 0.3 mW, 0.6 mW, 1.4 mW, and 3.4 mW.

### 2.8. SEM and SEM EDXS Analysis

SEM and SEM EDXS were carried out using a field emission SEM (FE-SEM) Zeiss Supra TM 35 VP (Carl Zeiss AG, Oberkochen, Germany), equipped with an EDX spectrometer silicon drift detector (SDD) EDX Ultim Max 100 (Oxford Instruments, Oxford, UK). TiON/C-CNF1 and TiON/C-CNF2 samples were analysed.

### 2.9. XRD

X-ray diffractograms of the sample TiON/C-CNF1 were recorded with a PANalytical X’Pert PRO MPD X-ray powder diffractometer (PANalytical B.V., Almelo, the Netherlands), with a radiation wavelength Cu Kα1 = 1.5406 Å in the α1 configuration with a Johansson monochromator on the primary side. The diffractograms were recorded with 0.034° resolution and 100 s signal integration time in the 2θ range from 20° to 60°.

### 2.10. TGA

TGA of TiON/C-CNF2 fabric was performed in NETZSCH, model STA 409 C Simultaneous Thermal Analyser (NETZSCH-Gerätebau GmbH, Selb, Germany). Analysis was performed in a Pt crucible in an oxygen atmosphere, with a heating rate of 10 °C/min and maximum temperature of 1100 °C.

### 2.11. CHNS Analysis

CHNS elemental analysis of TiON/C-CNF2 fabric was carried out with a PerkinElmer^®^ 2400 Series II CHNS/O Elemental Analyzer (PerkinElmer inc., Waltham, MA, USA).

### 2.12. Programs Used

The dimensions of the TiON/C-CNF1s and TiON/C-CNF2s were determined in the Fiji (ImageJ) [32] program (version 1.53o, Wayne Rasband and contributors, National Institutes of Health, USA) from calibrated micrographs. The nanoparticle size distribution (as equivalent radii) of TiON/C-CNF2s was determined by a custom-made program. XRD diffractogram was generated using CrystalDiffract^®^, a powder diffraction program for Mac and Windows (version 9.1.4(633), David Palmer, CrystalMaker Software Ltd., Oxford, England, www.crystalmaker.com, access date 23 June 2022) and EDXS spectrum was simulated with the DTSA-II program [33] (version Kelvin 2018-06-01, Nicholas W. M. Ritchie, Microanalysis Group, National Institute of Standards and Technology, Gaithersburg, MD, USA). At. % from STEM EDXS spectra were quantified using the Jeol AnalysisStation^®^ program (version 3, 8, 0, 34, JEOL Ltd., Tokyo, Japan). At. % of elements obtained from EELS were calculated with Gatan Digital Micrograph program (version 2.32.888.0, Gatan inc., Pleasanton, CA, USA).

## 3. Results and Discussion

### 3.1. Micro- and Nanostructure of TiON/C-CNFs

The SEM images of the microstructure of the TiON/C-CNF2 fabric at different magnifications can be observed in Figure 1. The nanofibers were spun in a continuous fashion and are intertwined and disordered. The average diameter of the TiON/C-CNF1s is 220 nm ± 30 nm and of the TiON/C-CNF2s 130 nm ± 60 nm. It was calculated from the diameters of 50 nanofibers measured manually from an SEM image (Appendix A). The diameter size distribution can be observed in Figure 2. Most TiON/C-CNF1s have diameters that are between 200 nm and 250 nm and only a few have diameters smaller than 150 nm or bigger than 250 nm. The low diameter dispersion is more suitable for the comparison to the theoretical model (this will be discussed later on). Most TiON/C-CNF2s have diameters that are between 50 nm and 150 nm and the number of TiON/C-CNF2s with bigger diameters decreases gradually. Very few TiON/C-CNF2s have a diameter below 50 nm. The higher diameter dispersion with more very small and very big diameters is more suitable for TEM analyses and single nanofiber electrical conductivity measurements. The difference in diameter dispersion can also be observed in Appendix A. At these magnifications, the TiON/C-CNF2 surface seems smooth and uniform, similarly to what was reported in other researches [20]. Higher magnification SEM images of TiON/C-CNF1s (Appendix A) show a rough surface, which is probably due to nanopores on the surface.

When we look at TiON/C-CNF2s with STEM (Figure 3), we can observe that the TiON/C-CNF2s consist of nanosized TiON particles, with an average size of 2.4 nm ± 0.8 nm embedded in an amorphous carbon matrix. The average size was calculated from 250 nanoparticles measured manually from a TEM image (Appendix A). The particle size distribution can be observed in Figure 2. Most TiON nanoparticles have diameters between 1 nm and 4 nm. The TiON nanoparticles are embedded in a lower density matrix in thicker TiON/C-CNF2s (diameter approximately 300 nm, Figure 3d,e,g,h). The matrix is less visible in thinner TiON/C-CNF2s (diameter approximately 50 nm, Figure 3a,b). This could be due to a higher overlap of TiON nanoparticles or due to a lower percentage of the matrix present. The darker spots observed in Figure 3b,g,h are nanopores, which can be more clearly observed in line profiles (Figure 3c,i). The average diameter of the nanopores is 6 nm ± 1 nm (Appendix A). The nanopores have much lower intensity compared to the rest of the TiON/C-CNF2. They form during nitridation when the organic material decomposes and forms escape routes for the gases. The TiON/C-CNF fabric also shrinks during nitridation (Appendix A). The shrinkage is estimated to be 40%.

### 3.2. Chemical Composition and Crystal Structure of TiON/C-CNFs

The samples were additionally analysed with XRD, SAED, CHNS elemental analysis, TGA, SEM EDXS, STEM EDXS, EELS, EELS mapping, and Raman to better characterise the composition and structure of TiON/C-CNFs. In Figure 4a (red line), an XRD diffractogram of TiON/C-CNF1 fabric can be observed, which confirms the presence of the TiN/TiON face centered cubic unit cell (FCC) nanocrystalline structure corresponding to the 1–4 nm sized nanoparticles of TiON/C-CNF2 observed in the STEM images (Figure 3). The size of crystallites was estimated with the Scherrer equation [34] and was found to be 8 nm on average from the two peaks. An XRD diffractogram was also simulated (Figure 4a, blue line), with a nanoparticle size of 6 nm (4.1987 Å, F m −3 m, Ti_0.7_(N_0.33_O_0.67_) based on the crystal structure from Inorganic Crystal Structure Database (ICSD) 426340 / Cambridge Crystallographic Data Centre (CCDC) 908784 [35], N and O occupancy changed to N_0.365_O_0.635_). The higher estimated nanoparticle sizes from XRD data compared to the nanoparticle sizes in STEM images can be explained by the possibility of bigger TiON nanoparticles hiding in the cores of thicker TiON/C-CNFs. The SAED pattern in Figure 4 c of the TiON/C-CNF2 piece (30 nm diameter) in Figure 4b confirms the presence of randomly oriented TiON nanoparticles or nanocrystallites. Raman spectra (Appendix A) also suggest the formation of TiON (details on Raman spectroscopy of TiON/C-CNF2s can be found in Appendix A).

The SEM EDXS spectrum of the TiON/C-CNF2 fabric in Appendix A shows the presence of carbon at around 10 at. % or 5 wt.% (STEM EDXS spectrum of TiON/C-CNF2 in Figure 5a and TGA and CHNS analysis of TiON/C-CNF2 fabric confirm the amount of carbon, details for TGA and CHNS analysis can be found in Appendix A), which can explain the lower density matrix in which the TiON nanoparticles are embedded. This is nicely observed in the STEM image of the TiON/C-CNF2 cross-section in Figure 3. Independently, the Raman spectra of the TiON/C-CNF2 fabric confirm the presence of carbon in the samples (Appendix A).

The N/O at. ratio of the TiON/C-CNF2 fabric calculated from SEM EDXS is 1.67 and is comparable to the N/O at. ratio calculated from the TGA and CHNS analysis of the TiON/C-CNF2 fabric, which is 1.65. The average N/O at. ratio calculated from the STEM EDXS measurements of several TiON/C-CNF2s is 1 ± 0.2. The STEM EDXS spectrum was also simulated (Appendix A) and the simulated spectrum is in agreement with the automatic quantification of the STEM EDXS program (details about the STEM EDXS simulation can be found in Appendix A). The average N/O at. ratio calculated from the EELS measurements of several TiON/C-CNF2s is 0.6 ± 0.05 (details on N/O at. ratio determination with EELS can be found in Appendix A). Different results from these two methods are expected because the SEM EDXS and TGA and CHNS analyses measure the bulk N/O at. ratio of the TiON fabric, while EELS and STEM EDXS measure the local N/O at. ratio of individual TiON/C-CNF2s that have a diameter of around 50 nm or less.

The comparison of SEM EDXS spectra of TiON/C-CNF1 and TiON/C-CNF2 fabric can be observed in Appendix A. Due to the pursuit of fabricating samples with different diameter distributions, which were required in order to successfully analyse the samples further with other techniques, slight differences in composition could not be avoided. The comparison of compositions can be found in Table 1.

EELS mapping of an individual TiON/C-CNF2 in Figure 6c,d shows a thin layer of lower N/O at. ratio on the surface and in the thinner parts of the TiON/C-CNF2. This can be more clearly observed in the line profiles in Figure 6e,f. In Figure 6e, the N (red) line dominates the O (green) line throughout the core of the TiON/C-CNF2 and is only surpassed by the O line on the outermost parts (5 nm—surface of the nanofiber) and in pores where the TiON/C-CNF2 is thinner. Similarly, the N/O at. ratio in Figure 6f decreases quickly on the surface and is also lower in nanopores.

The same phenomenon can be observed from the EELS mapping of the cross-section of a 300 nm thick single TiON/C-CNF2 (Figure 7). The thin surface layer has a lower N/O at. Ratio, which increases towards the inner parts of the TiON/C-CNF2 (Figure 7c,d). This is obvious from the line profiles in Figure 7e,f. In Figure 7e, the O line is higher than the N line only at the surface and the N/O at. ratio in Figure 7f is steadily increasing from 0.5 on the surface to around 2 towards the core. It is important to note that the thickness of the cross-section of the 300 nm thick TiON/C-CNF2 is increasing from the surface to the centre (right to left side of each profile). It was cut under an angle that ensures that some area of the cross-section would be thin enough to be analysed with EELS and EELS mapping. In Figure 7d, it can be observed that the N/O at. ratio quickly starts diverging (very high positive and negative values) when approaching the inner part of the TiON/C-CNF2. Due to the increasing thickness, the background in EELS measurements is increasing as well and the N and O edges are becoming hidden in the background and the N/O at. ratio calculation makes less sense. This can also be observed in the line profile in Figure 7e, where the edge integration value of N and O is decreasing and overlapping more closer to the core. The line profile of the N/O at. ratio in Figure 7f was, therefore, restricted to only 65 nm from the surface.

It can be found in the literature that the surface of nano-scale TiN and TiON can already be oxidised at room temperature [8]. Initially, N atoms in TiN or TiON are exchanged for O atoms and in the later stages of oxidation, TiO_2_ can also be formed [7]. This also explains the higher amounts of O on the surface, and consequently, the different results of N/O at. ratio from SEM EDXS and EELS. We can expect a lower N/O at. ratio (relatively more oxygen) with EELS because EELS can only be carried out on thin regions of the sample, where the surface is a higher percentage of the analysed material. The SEM EDXS measurements can obtain information from the core of thicker samples where more N is present, leading to higher N/O at. ratios.

### 3.3. Electrical Conductivity and Electrochemical Measurements

Individual TiON/C-CNF2s were put on Protochips electrical chips and connected to the four chips’ contacts with Ga ions deposited platinum with FIB-SEM in a four-probe electric circuit configuration, as can be observed in Figure 8. The four-probe electrical resistance was measured inside the TEM vacuum with the Protochips Aduro^TM^ power supply. To avoid interaction with the conductivity measurement, the electron beam was switched off. In total, four TiON/C-CNF2s were measured successfully and their resistivity was calculated based on the measured and calculated dimensions using the following formula:(1)$$\rho =R\frac{A}{l}=R\frac{\pi r^{2}}{l},$$
where ρ is resistivity, R is resistance, r is the average nanofiber radius and l is the nanofiber length. It is assumed that the TiON/C-CNF2s are cylindrical. The average nanofiber radius was calculated from many manual diameter measurements along the TiON/C-CNF2. The measurements and calculations are listed in Table 2. The average conductivity of the four TiON/C-CNF2s measured is 1.2 kS/m ± 0.5 kS/m (average resistivity 1.0 mΩm ± 0.5 mΩm), which is in the range of carbon nanofibers [27,36,37], the type of support that is supposed to be replaced with TiON. The electrical conductivity of similar TiON nanofibers reported elsewhere in the literature can be in the same order [7] or up to around 30× higher [7] and in some cases, even around 3000× higher [8]. The fact that TiON nanoparticles are immersed in a carbon matrix can explain the conductivity, compared to some more conductive TiON nanofibers from other research studies [7,8]. TiN nanofibers can have around 5× higher conductivity and bulk TiN around 400× higher [26] compared to our TiON/C-CNF2s. The TiON/C-CNF2s exhibit ohmic behaviour; a typical current–voltage (I–V) curve can be observed in Figure 9.

Sheet resistivity of TiON/C-CNF1 fabric was also measured. For a typical sample, we measured the sheet resistivity of around 630 Ω□ and nanofiber radius of 130 nm. Together with the thickness measurement of 30 μm (Appendix A), bulk conductivity was calculated to be 0.053 kS/m (bulk resistivity 19 mΩm), which is 4× lower than a similar fabric from the literature [19]. By comparing bulk resistivity of the TiON/C-CNF1 fabric and resistivity of a single TiON/C-CNF2, we can determine the characteristic length between nanofiber contacts, which is a measure of mesh density. We do this by approximating the nanostructure of the TiON/C-CNF1 fabric sample as a mesh of resistors. SEM images of the contacts between the nanofibers (Appendix A) confirm that contacts are robust and are not expected to contribute significant contact resistances. This theoretical model can be generalised from a simple regular cubic lattice of resistors to a random anisotropic mesh. A cubic lattice of resistors can be shown to have specific resistivity equal to $\rho_{bulk}=Rl$ in all directions, where *R* and *l* are resistance and length of each lattice segment. Expressing both values in terms of nanowire cross-section A and resistivity of material ρ allows us to obtain an approximation of the characteristic distance between nanofiber contacts, using the following equation:(2)$$l=\sqrt{\frac{A\rho_{bulk}}{\rho }}$$
In this case, it is approximately 1.0 µm and is in agreement with the characteristic distance between the nanofiber contacts estimated from the SEM images (1–3 µm, Appendix A). It should be noted that the samples are truly anisotropic only in X and Y dimensions.

Specific capacitance of TiON/C-CNF1 fabric was determined with charge–discharge curves at different current densities (4–60 Ag^−1^) and with EIS, as shown in Figure 10. Methods for determination were found in the literature [16,38]. Coulombic efficiencies for the charge–discharge curves indicate that the efficiency is higher when the current is higher (73% at 4 A/g, and 93% at 60 A/g). The measured capacitances, 458 F/g, 708 F/g and 1300 F/g from the CV, charge–discharge and EIS, respectively, are comparable to the best performing Ti-based materials in the literature [17]. The material has high capacitance, most likely due to the combination of (a) its morphology, which gives it a high surface area, and (b) the great conductivity of electrospun nanowires. This shows promise in the supercapacitor field.

### 3.4. Estimation of Electrical Conductivity of TiON/C-CNF2s

The electrical conductivity of TiON/C-CNF2s can also be estimated from the material characterisation data. The TiON/C-CNF2s are a three-phase composite material made from TiON nanoparticles embedded in a C matrix with nanopores. TiON and C phases are conductive and the nanopores have a conductivity of 0 kS/m. Based on the shape of the C edge in EELS data (Appendix A), we can determine the type of C in the C matrix [39]. There is no white line visible at around 285 eV, characteristic for σ bonds [39], which means that the C matrix consists of amorphous C. The electrical resistivity of amorphous C is in the range of 0.1 mΩm−0.7 mΩm (1.4 kS/m–10 kS/m) [40]. The electrical conductivity of TiON depends on the N/O at. ratio. The measured ratios from different measurement techniques range from 0.6 for EELS, 1 for STEM EDXS, 1.65 for TGA and CHNS, to 1.7 for SEM EDXS. Moreover, the electrical conductivities for these N/O at. ratios reported in the literature vary widely due to different synthesis methods [5,7,41,42,43], which gives us an electrical conductivity range for TiON nanoparticles between 0.07 kS/m and 50 kS/m. The electrical conductivities of amorphous C matrix and TiON nanoparticles might not be comparable and conductivities of both are definitely not comparable to the conductivity of nanopores; therefore, a simple rule of mixture (ROM) cannot be used for effective conductivity determination of the TiON/C-CNF2 sample. The effective medium theory (EMT) for a three-phase composite (spherical inclusions of TiON and nanopores in a C matrix) in 3D is more suitable for this material [44].
(3)$$f_{1}\frac{\sigma_{1}-\sigma_{eff}}{\sigma_{1}+2\sigma_{eff}}+f_{2}\frac{\sigma_{2}-\sigma_{eff}}{\sigma_{2}+2\sigma_{eff}}+f_{3}\frac{\sigma_{3}-\sigma_{eff}}{\sigma_{3}+2\sigma_{eff}}=0,$$
where $f_{i}$ is the volume fraction of phase i, $\sigma_{i}$ is the electrical conductivity of phase i and $\sigma_{eff}$ is the effective conductivity of the composite. Based on the images in Appendix A, the volume fraction was calculated to be 40 vol.% for the TiON nanoparticles, 50 vol.% for C matrix, and 10 vol.% for nanopores. It is assumed that the surface area fraction from the 2D image is equal to the volume fraction. The solved equation gives us an effective conductivity range from 0.45 kS/m to 19 kS/m for our TiON/C-CNF2s. Note that the measured electrical conductivities of single TiON/C-CNF2s (Table 2) are within this range.

## 4. Conclusions

Electrospun TiON/C-CNFs, thermally treated in ammonia, were thoroughly analysed using various characterisation methods and their electrical conductivity was measured. It was found that the 50 nm–150 nm thick TiON/C-CNF2s consist of around 2.4 nm sized TiON nanoparticles embedded in an amorphous carbon matrix. Up to 10 vol.% of pores were also present in the material. The N/O at. ratio ranged from 0.6 for EELS, 1 for STEM EDXS, 1.65 for TGA and CHNS, to 1.7 for SEM EDXS, and EELS mapping showed a lower N/O at. ratio on the surface of the TiON/C-CNF2s. Using in situ four-probe electrical conductivity measurements, it was found that the conductivity is better than some carbon-based nanofibers, which confirms that TiON can be seen as a good alternative support to carbon-based supports for electrocatalysts and also other electrochemical applications. The measured capacitance is comparable to the best performing Ti-based materials in the literature. The estimated mesh density of the TiON/C-CNF fabric from the electrical measurements is in agreement with the SEM observations. Lastly, the electrical conductivity of a single TiON/C-CNF2 is within the range of electrical conductivity estimation based on material characterisation.

## Figures and Tables

**Figure 1 nanomaterials-12-02177-f001:**
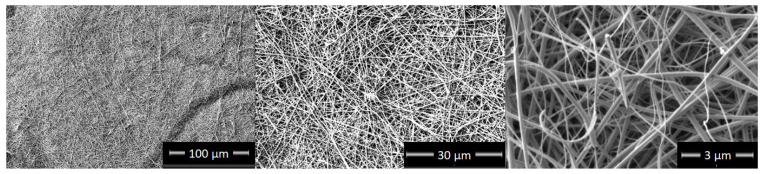
Scanning electron microscopy (SEM) images of Titanium oxynitride carbon composite nanofibers (2nd sample, TiON/C-CNF2) fabric at different magnifications.

**Figure 2 nanomaterials-12-02177-f002:**
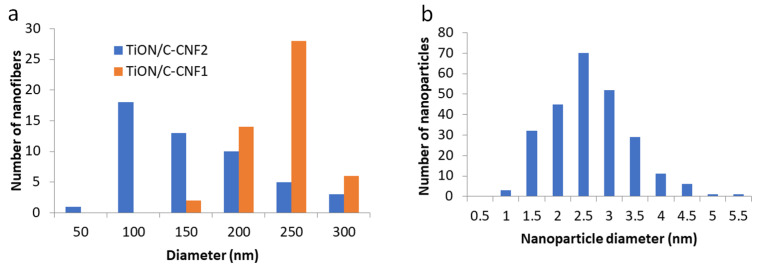
Diameter size distribution of 50 Titanium oxynitride carbon composite nanofibers (1st sample, TiON/C-CNF1s) and TiON/C-CNF2s (**a**) and particle size distribution of 250 nanoparticles of TiON/C-CNF2 sample (**b**). Distributions were determined from images in Appendix A.

**Figure 3 nanomaterials-12-02177-f003:**
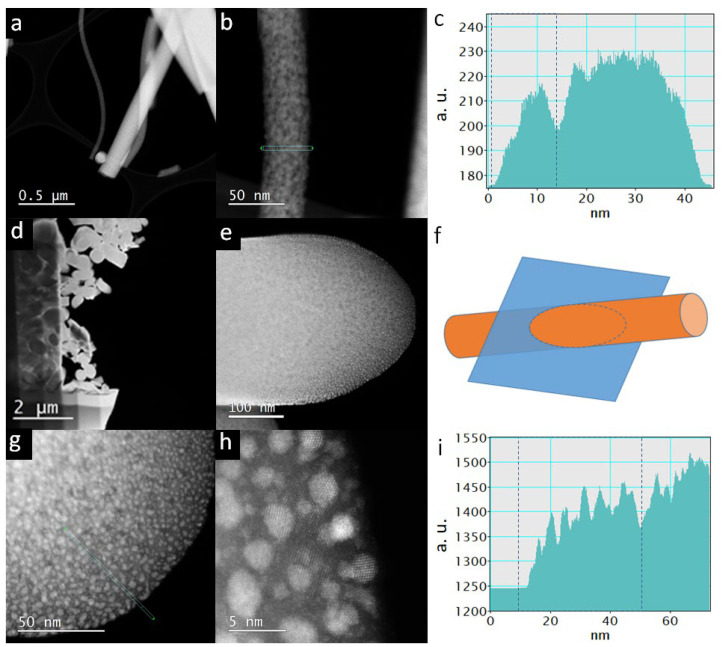
(**a**,**b**) scanning transmission electron microscopy (STEM) high angle annular dark field (STEM-HAADF) images of TiON/C-CNF2s, (**c**) line profile of the image b with an integrated width of 3.5 nm (marked with green dots and turquoise lines), (**d**,**e**,**g**,**h**) lamella of cross-section of TiON/C-CNF2s, (**f**) scheme of how the TiON/C-CNF2 was cut, (**i**) line profile of the image g with an integrated width of 1.5 nm (marked with green dots and turquoise lines).

**Figure 4 nanomaterials-12-02177-f004:**
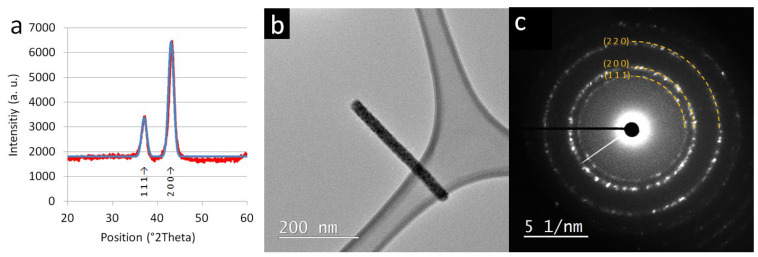
(**a**) (X-ray diffraction) XRD diffractogram of the TiON/C-CNF1 fabric (red) and a simulated XRD diffractogram of titanium oxynitride (TiON) nanoparticles (blue, 6 nm, based on the crystal structure Cambridge Crystallographic Data Centre (CCDC) 908784, N and O occupancy changed from N33 O67 to N365 O635), (**b**) transmission electron microscopy (TEM) image of a TiON/C-CNF2 segment (30 nm diameter), (**c**) selected area electron diffraction (SAED) pattern of the TiON/C-CNF2.

**Figure 5 nanomaterials-12-02177-f005:**
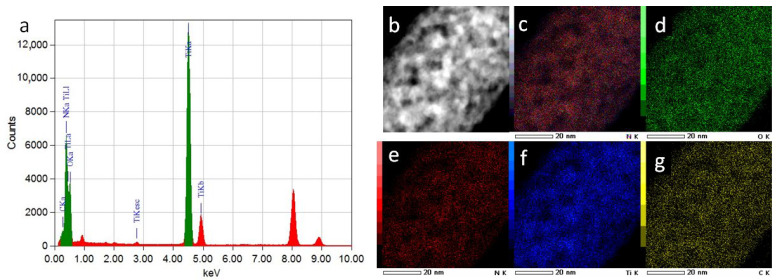
(**a**) STEM energy dispersive X-ray spectroscopy (EDXS) spectrum of the TiON/C-CNF2, (**b**) STEM-HAADF image of the TiON/C-CNF2 location for STEM EDXS analysis, (**c**) overlay image of images (**d**–**f**), (**d**) O elemental mapping image, (**e**) N elemental mapping image, (**f**) Ti elemental mapping image, (**g**) C elemental mapping image.

**Figure 6 nanomaterials-12-02177-f006:**
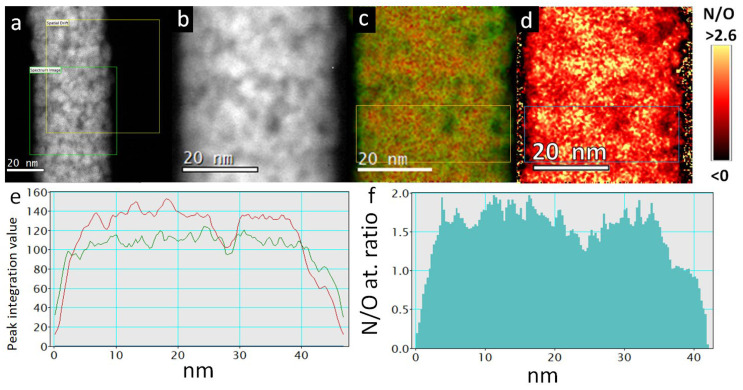
Electron energy loss spectroscopy (EELS) mapping of a TiON/C-CNF2. (**a**) STEM-HAADF image of the chosen locations for EELS mapping (in green) and spatial drift monitoring (in yellow), (**b**) the mapped area with EELS mapping (each pixel is a whole EELS spectrum), (**c**) N and O edges extracted EELS mapping image with more nitrogen content in red and more oxygen content in green (smoothed image), (**d**) N/O at. ratio in every pixel as a colour (smoothed image), (**e**) a line profile of the image (**c**) with an integrated width 15 nm (marked with an orange square on image (**c**)), where the red line represents N and the green line represents O edge integration value, (**f**) a line profile of the image (**c**) with an integrated width of 15 nm (marked with a blue square on image (**d**)).

**Figure 7 nanomaterials-12-02177-f007:**
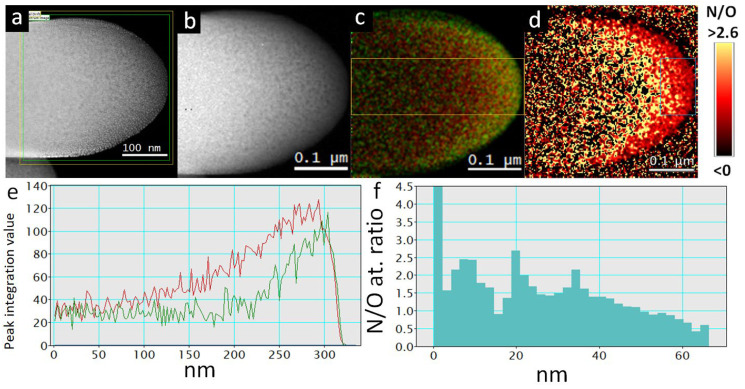
EELS mapping of a cross-section of a TiON/C-CNF2. (**a**) STEM-HAADF image of the chosen locations for EELS mapping (in green) and spatial drift monitoring (in yellow), (**b**) the mapped area with EELS mapping (each pixel is a whole EELS spectrum), (**c**) N and O edges extracted EELS mapping image with more nitrogen content in red and more oxygen content in green (smoothed image), (**d**) N/O at. ratio in every pixel as a colour (smoothed image), (**e**) a line profile of the image c with an integrated width of 100 nm (marked with an orange square on image (**c**)), where the red line represents N and the green line represents O edge integration value, (**f**) a line profile of the image c with an integrated width of 100 nm (marked with a blue square on image (**d**)).

**Figure 8 nanomaterials-12-02177-f008:**
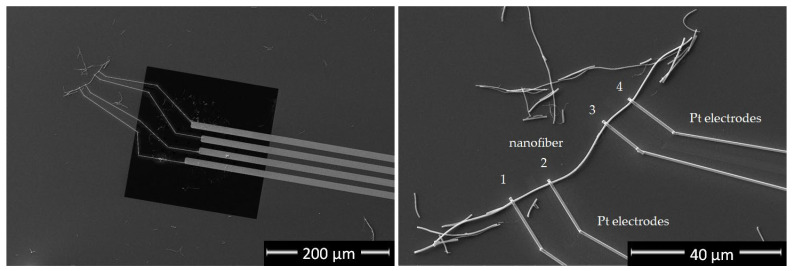
SEM image of a single TiON/C-CNF2 with four-probe contact on a Protochips electrical chip.

**Figure 9 nanomaterials-12-02177-f009:**
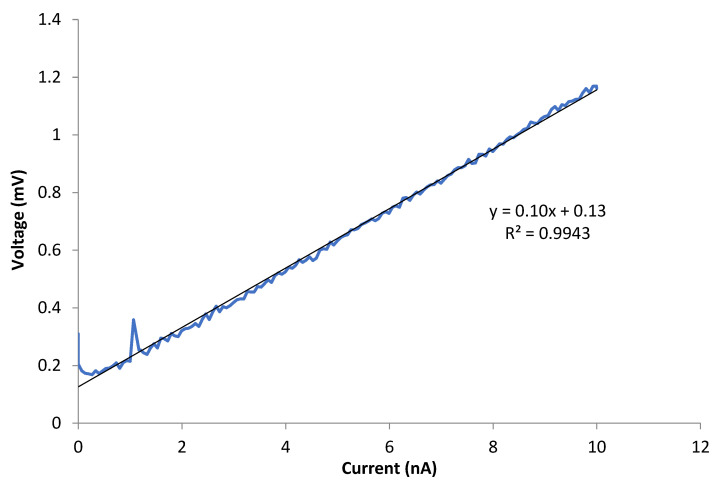
Typical current-voltage (I–V) curve of single TiON/C-CNF2 exhibiting ohmic-type conductivity.

**Figure 10 nanomaterials-12-02177-f010:**
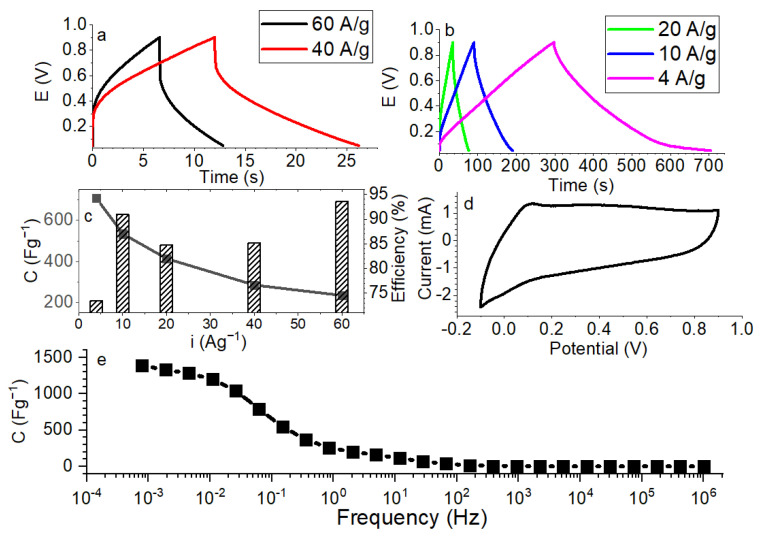
(**a**) Charge–discharge curves of TiON/C-CNF1 fabric at 40 Ag^−1^ and 60 Ag^−1^, from 0.05 V to 0.9 V, (**b**) charge–discharge curves of TiON/C-CNF1 fabric at 4 Ag^−1^, 10 Ag^−1^, and 20 Ag^−1^, from 0.05 V to 0.9 V, (**c**) specific capacitances and coulombic efficiency of TiON/C-CNF1 fabric for different current densities from charge–discharge curves, (**d**) cyclic voltammetry (CV) at a scan rate of 10 mV/s, and (**e**) specific capacitance of TiON/C-CNF1 fabric determined with electrochemical impedance spectroscopy (EIS).

**Table 1 nanomaterials-12-02177-t001:** Composition of samples TiON/C-CNF1 and TiON/C-CNF2. The composition was obtained from SEM EDXS spectra in Appendix A.

Sample	At.%		Wt.%		N/O at. Ratio	n(Ti):n(C)	m(TiON):m(C)
TiON/C-CNF1	Ti	36.8	Ti	65.6	1.13	4.84	28.4
	O	26.1	O	15.6			
	N	29.5	N	15.4			
	C	7.6	C	3.4			
TiON/C-CNF2	Ti	33.8	Ti	63.2	1.67	2.66	6.87
	O	20.0	O	12.5			
	N	33.4	N	18.3			
	C	12.7	C	6.0			

**Table 2 nanomaterials-12-02177-t002:** Table of resistivity, dimensions and specific resistivity of single TiON/C-CNF2s.

Nanofiber Sample No.	Resistance R (kΩ)	Average Nanofiber Radius r (nm)	Nanofiber Length L (µm)	Resistivity Ro (mΩm)	Conductivity (kS/m)
1	100	207	22.3	0.6	1.7
2	120	230	35.2	0.6	1.8
3	872	170	44.5	1.8	0.6
4	130	230	18.0	1.2	0.8
Average				1.0 ± 0.5	1.2 ± 0.5

## Data Availability

The data presented in this study are available upon reasonable request from the corresponding author.

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
