# Peer review of "Microstructure and Electrical Conductivity of Electrospun Titanium Oxynitride Carbon Composite Nanofibers"

_nanomaterials, 2022, doi:10.3390/nano12132177_

Round 1

Reviewer 1 Report

This paper is interesting, well-argued and fits well with the scopes of the journal.

By focusing on the realization of nanofibers based on Titanium Oxynitride, it analyses the electrical and electrochemical properties, the morphology, the chemical composition and structure of the fibers.

Data are clearly presented, and conclusions are consistent with both premises and discussion.

I have only minor revisions:

-        Pg 3 line 117 and 130: I suggest expressing the composition of fibers in wt% because it could be more easy to underline the differences. However, I do not understand why two different procedures were used and which are the differences between the two procedures, except for the exact composition of the electrospinning solution and the elecrospinning parameters.

-        Pg 4 Line 161 and 167: the thickness of the samples is missing

-        Pg 5 Line 236: is the diameter the same for the nanofibers obtained with both procedures?

Reviewer 2 Report

Nanowires of TiON is synthesised via electrospinning method. The paper is not organized well, and it should be checked by the authors and resubmit it again. The following comments should be considered:  

1.       All Figure words is replaced by “Error! Reference source not found.”

2.       A comprehensive compression of this TiON with the other reported TiON studies should be added and explained.

3.       Fig. 1: SEM images are belong to which samples, 50TiON or 250 TiON?

4.       The “TiON nanowire” should be “TiON nanowire composite”, as the nanowires are not pure TiON, but they having carbon also.

5.       High magnifications SEM images about 200 nm in the scale bare should be provided for the samples, to see the nanowire composite sizes clearly.

6.       What is 50 TiON and 250 TiON?

7.       How the authors measured the nanowire composite dimeters in Fig. 2?

8.       Fig. 5: channel A and B in the figure should be removed.

9.       Fig.6a: the charge and discharge curves of 40 and 60 A/g should be separated and showed in another graph with clear view.

10.   Fig.6b: the corresponding coulombic efficiency should be added.

11.   Some cycling data for this material should also be provided.

12.   Fig. 6 should be shifted to the end of manuscript after Fig. 10.

13.   How much is areal mass loading for supercap application?

14.    How much is the ratio of carbon to TiON in the nanowire composite? Should be analysed.

Round 2

Reviewer 2 Report

The paper can be accepted in this form.